# Contribution of Physical Activity to the Oxidative and Antioxidant Potential in 60–65-Year-Old Seniors

**DOI:** 10.3390/antiox12061200

**Published:** 2023-05-31

**Authors:** Bartłomiej K. Sołtysik, Kamil Karolczak, Tomasz Kostka, Serena S. Stephenson, Cezary Watala, Joanna Kostka

**Affiliations:** 1Department of Geriatrics, Medical University of Lodz, Haller Square No. 1, 90-419 Łódź, Poland; 2Department of Hemostatic Disorders, Medical University of Lodz, Mazowiecka Street 6/8, 92-215 Łódź, Poland; 3Department of Gerontology, Medical University of Lodz, Milionowa Street No. 14, 93-113 Łódź, Poland

**Keywords:** lipid peroxides, free thiol groups, free amino groups, superoxide, antioxidant status, physical activity, elderly

## Abstract

Both acute exercise and regular physical activity (PA) are directly related to the redox system. However, at present, there are data suggesting both positive and negative relationships between the PA and oxidation. In addition, there is a limited number of publications differentiating the relationships between PA and numerous markers of plasma and platelets targets for the oxidative stress. In this study, in a population of 300 participants from central Poland (covering the age range between 60 and 65 years), PA was assessed as regards energy expenditure (PA-EE) and health-related behaviors (PA-HRB). Total antioxidant potential (TAS), total oxidative stress (TOS) and several other markers of an oxidative stress, monitored in platelet and plasma lipids and proteins, were then determined. The association of PA with oxidative stress was determined taking into the account basic confounders, such as age, sex and the set of the relevant cardiometabolic factors. In simple correlations, platelet lipid peroxides, free thiol and amino groups of platelet proteins, as well as the generation of superoxide anion radical, were inversely related with PA-EE. In multivariate analyses, apart from other cardiometabolic factors, a significant positive impact of PA-HRB was revealed for TOS (inverse relationship), while in the case of PA-EE, the effect was found to be positive (inverse association) for lipid peroxides and superoxide anion but negative (lower concentration) for free thiol and free amino groups in platelets proteins. Therefore, the impact of PA may be different on oxidative stress markers in platelets as compared to plasma proteins and also dissimilar on platelet lipids and proteins. These associations are more visible for platelets than plasma markers. For lipid oxidation, PA seems to have protective effect. In the case of platelets proteins, PA tends to act as pro-oxidative factor.

## 1. Introduction

The pathogenesis of an increasing number of diseases, including cardiometabolic disorders, is seen in oxidative stress [1]. One of the factors that raise doubts about its effect on the oxidative potential is exercise and physical activity (PA) [2]. According to various studies, PA increases the total antioxidant potential (TAS) or, conversely, generates increased total plasma oxidative status (TOS) [3,4]. Both prolonged or short-duration high intensity exercise results in an increased production of reactive oxygen species (ROS) in active skeletal muscles, which results in the formation of oxidized lipids and proteins in the working muscles [3]. The formation of ROS is a natural product of muscle work during PA. On the other hand, this phenomenon may increase TAS [5]. The beneficial effects of physical activity on antioxidant status and oxidative stress, especially in aged subjects, have been reported in some studies [4,6,7,8]. Notwithstanding, if ROS generation reduces the TAS, oxidative stress appears, which further leads to molecular and cellular damage and consequently to various disorders [9]. Hence, it is important to explore more closely different aspects of PA as the possible trigger of oxygen metabolism pathways in regard to cardiovascular outcomes. Such an assessment seems to be especially important in aging subjects with increased cardiovascular risk.

TAS is a laboratory marker of the body’s potential against oxidation, a measure of the antioxidant reserve that inhibits oxidation processes. In addition to well-known endogenous and dietetic antioxidants with such an effect, statins, angiotensin converting enzyme inhibitors, angiotensin II receptor blockers (ARB) [10] and regular PA [11,12,13] have been reported as potential modulators of TAS. Regular PA boosts the expression of antioxidant enzymes, decreases pro-oxidant ones, which enhances the TAS [14]. It seems that regular PA may affect endothelium and platelet in a different way, which complicates the assessment of impact of PA on cardiovascular risk [9,15,16].

Oxidation disrupts the functioning of the cell proteins, directing inevitably toward apoptosis. Superoxide radical, hydrogen peroxide or hydroxide radical oxidize the protein and membrane thiol groups (-SH), which results in biological inactivation of proteins, and disintegration of cell membrane [17]. In the currently available literature, there are studies showing a positive correlation between lipid peroxidation and the level of PA [18], as well as other studies indicating the opposite relation [19]. The discussed studies, however, concern various protocols for the assessment of PA, various groups in terms of sex, age and medical history [20,21,22,23].

The concentration of both free thiol and amino groups of plasma proteins and platelets can be considered an indicator of damage to protein structures by ROS and thus indirectly indicates the level of oxidative stress acting on these structures [24,25,26]. However, there is a paucity of studies assessing the relationship between the concentration of free thiol and amino groups and PA, and the results are inconclusive [25,27,28,29,30]. Likewise, the effect of PA on the production of the superoxide anion appears to be ambiguous, as it is in the case of other reactive oxygen species. Moreover, currently available literature includes mainly studies in young athletes [31,32].

In previous studies, we have reported the complex associations between oxidative/antioxidative potential and cardiometabolic risk profile, as well as between PA and cardiovascular risk factors in the population of 60–65-year-old subjects [33,34]. In this study, we estimate how the PA (assessed both through the profile of health behaviors and overall energy expenditure) may relate to the TAS, TOS and the variety of plasma and platelets markers of an oxidation in the population of younger seniors.

## 2. Materials and Methods

The study presents the results from the project “The occurrence of oxidative stress and selected factors for cardiovascular risk and functional status of older people in the context of workload” (Central Institute For Labor Protection-National Research Institute, Warsaw, Poland). The study included 150 women and 150 men, age-matched during the recruitment process. All the participants were independent community-dwelling volunteers. The inclusion criteria were age within the range of 60 to 65 years and the consent to participate in the study. The study was approved by the Committee on the Ethics of Research in Human Experimentation at the Medical University of Lodz (RNN/648/14/KB dated 23 September 2014), and was performed according to the guidelines of the Helsinki Declaration for human research [33,34].

Three hundred sex-matched subjects were qualified to the study group. The assessment of anthropometric checkup, sociodemographic data, medical anamnesis and medication intake was executed in the Department of Geriatrics at the Medical University of Lodz. Weight and height, waist and hip circumference were registered, which allowed for determination of body mass index (BMI), waist–hip ratio (WHR) and waist–height ratio (WHTR). The estimation of metabolic syndrome was conducted according to the guidelines of International Diabetes Federation [35]. Obesity was diagnosed with BMI ≥ 30 kg/m^2^.

### 2.1. Physical Activity

The PA was assessed on the basis of the Seven-Day Recall Physical Activity Questionnaire [36] and the Stanford Usual Activity Questionnaire [37]. Both questionnaires have been previously described in detail [38].

The Seven-Day Recall PA Questionnaire assesses overall energy expenditure through analyzing PA during the previous seven days. The method estimates the hours spent during sleeping, light (activities with energy expenditure of 1.5 kcal/min), moderate (activities with energy expenditure of 4 kcal/min), hard (activities with energy expenditure of 6 kcal/min), and very hard activities (activities with energy expenditure of 10 kcal/min) [36]. The Seven-Day Recall score (total energy expenditure over previous week presented in kcal·kg^−1^·day^−1^) was then determined as PA-energy expenditure (PA-EE).

The Stanford moderate index allows an evaluation of health-related PA behaviors of light and moderate intensity. The respondents indicate the type of behavior typical of their exercise habits: climbing the stairs instead of using the elevator, walking instead of driving for a short distance, parking the car further away from the destination in order to approach on foot, walking before or after lunch or dinner, exiting the bus or tram one stop before in order to walk the remaining distance, or performing other activities of a similar nature [37].

In the Stanford Hard index, the respondent specifies the following activities completed habitually during the last three months in weekly intervals for at least: jogging or running at least 10 miles, play vigorous racquet sports at least five hours, play other strenuous sports at least five hours, cycle at least 50 miles, swim at least two miles [38]. The Stanford Moderate (six habitual moderate activities; scoring points 0–6) and Hard (five habitual intensive activities; scoring points 0–5) indices were computed as a numerical sum of points for each activity and used for further comparisons. These two PA indices are presented as PA-health-related behaviors I (PA-HRB I) and PA-health-related behaviors II (PA-HRB II), respectively. Both questionaries have demonstrated their high validity in older individuals against double labelled water and have been assessed in the present study in accordance with standardized protocols [39,40].

### 2.2. Laboratory Measurements

The blood testing was conducted in the Department of Hemostatic Disorders at the Medical University of Lodz. The laboratory protocol comprises a wide range of variables. The concentration of glucose, total cholesterol (TC), low-density lipoprotein cholesterol (LDL-C), high-density lipoprotein cholesterol (HDL-C), triglycerides (TG), uric acid (UA) was assessed spectrophotometrically (DIRUI CS 400, Changchun, China). Homocysteine (Hcy) was measured using the analyzer Immulite 2000 XPi (Siemens, Erlargen, Germany). Plasma concentration of von Willebrand factor (vWf), soluble vascular cell adhesion protein 1 (VCAM-1), soluble intracellular adhesion molecule 1 (ICAM-1) were checked with commercial ELISA kits (Abcam, Cambridge, UK). Thromboxane B2 was generated under either static (TXB2s) or dynamic conditions (TXB2d), depending on the employed protocol) [41,42].

### 2.3. Estimation of TAS, TOS and of the Markers of the Oxidative Damage of Proteins and Lipids in Blood Platelets and Plasma

TAS and TOS were estimated with commercially accessible kits produced by LDN Labor Diagnostika Nord GmbH & Co. KG (Nordhorn, Germany). TAS level was established according to ABTS cation decay reaction [43]. Platelet and plasma lipid peroxides were assessed with the result of peroxide–peroxide reaction converting tetra-methyl benzene (TMB) to a colored product measured photometrically (450 nm) [44,45]. The concentration of free protein thiol groups in platelet lysate and in blood plasma was performed according to a previously published protocol [46]. Briefly, 300 µL of isolated blood plasma or platelet lysate (platelet lysate prepared from platelets isolated via differential centrifugation by freezing and thawing the sample three times at −80 °C and 37 °C) was added to 300 µL of 10% (*m*/*v*) SDS. Then, 2.4 mL of 10 mM phosphate buffer (pH 8.0) was added. The absorbance of the sample was measured at λ = 412 nm, giving A0. Then, 300 µL of Ellman’s reagent (5,5′-dithiobis-(2-nitrobenzoic acid, DTNB)) prepared in 10 mM phosphate buffer (pH 8.0) was added. In the case of control samples, 10 mM phosphate buffer (pH 8.9) was added without DTNB. The samples were incubated for one hour at 37 °C, after which the absorbance was read at λ = 412 nm, yielding A1. A0 was subtracted from A1 to obtain a measure of the thiol content of proteins. The concentration of free thiol groups in platelet and plasma proteins was calculated using the millimolar absorption coefficient at 412 nm for the thio-2-nitrobenzoic acid anion formed in the reaction equal to 13.6 mmol^−1^ × L × cm^−1^.

The concentration of free amino groups in platelet and plasma proteins was measured based on a previously published method [47]. The lysates of blood platelets and blood plasma were mixed with 2,4,6-trinitrobenzenesulfonic acid (TNBS, 0.01% solution prepared in 0.1 mol/L sodium bicarbonate buffer, pH 8.5) and stored in the dark for two hours at 37 °C. In the case of the control samples, sodium bicarbonate buffer was added with no TNBS. Next, 10% solution of SDS and 1 mol/L HCl were added to the samples. The concentration of the colored product was appraised on the basis of the absorbance measured at λ = 335 nm. Parallel calibration curves were prepared for lysine (presence of two amino groups in the molecule) and glutamic acid (presence of one amino group in the molecule). The unit of concentration of amino groups in the tested proteins was µg of lysine, which was calculated using the following relationship: A335Lys- A335Glu = µg Lys/mL.

For thiol and amino groups of proteins, the results are presented per µg or mg of protein. Protein concentration was determined via a standard BCA method. Platelets were isolated via differential centrifugation, as described in details elsewhere [48]. Platelet count in suspension was monitored according to Walkowiak et al. [49] and platelets were suspended in Tyrode’s buffer at a final count of 3 × 10^8^ platelets/mL. Superoxide anion generation by both non-homocysteinylated and in vitro homocysteinylated platelets was described as a product of spectrophotometric method of cytochrome c reduction by superoxide anion, described by Gresele et al. [41,50]. The reaction is based on the reduction of ferricytochrome c to ferrocytochrome c. The course of the reaction can be followed spectrophotometrically because the light absorption spectra of both forms of cytochrome, measured at 550 nm, differ. Since cytochrome c does not pass through the cell membrane, its reduction indicates the release of the anion radical outside the cells. It is therefore a spectrophotometric method that detects the superoxide anion radical indirectly [51,52,53].

### 2.4. Statistical Analysis

The statistical analysis of the variables was performed on the basis of parametric and non-parametric tests. The analysis of empirical distributions of the variables was established with the Shapiro–Wilk test. The homogeneity of variance was checked using the Levene test. The arithmetic mean (m) and the median (Me) were calculated. The standard deviation (SD) and the values of the lower and upper quartiles were adopted as the measures of the dispersion. The characteristics of the study population have been presented as comparisons between groups with PA-EE below and above median, calculated separately for women and men.

For nominal (qualitative) variables (e.g., sex), the χ2 test or the Fisher’s exact test (depending on subgroup sizes) were used for the analyses of contingency tables. For quantitative features, when the condition of normal distribution and variance equality (homogeneity) was met, the parametric Student’s *t*-test was used; otherwise, the non-parametric Mann–Whitney U test was utilized. To assess the relationship between two quantitative variables, Spearman’s rank correlation test (rho coefficient) was used.

Due to the floor effect, Stanford II data were dichotomized (at least one activity present; at least one activity absent) and such variables were presented as the percentages of subjects meeting the condition of at least 1 Stanford II activity. Multivariable analyses were performed with the use of a general linear model. Independent variables showing statistical significance in univariate tests (*p* < 0.05) entered the model. The quantitative variables which has not met the criterion of normal distribution were logarithmically transformed before inclusion in multivariate models. For all statistical analyses, the level of statistical significance was established as *p* < 0.05. Statistical analyses were performed using the Statistica software version 13.1 (StatSoft, Kraków, Poland).

## 3. Results

Median PA-EE in the whole studied population was 43.2 (39.1–50.4) kcal/kg/day, PA-HRB I was 3.0 (1.0–4.0), and 47 (15.6%) subjects reported at least one PA-HRB II activity. PA-EE was higher in women (44.2 vs. 42.0 kcal/kg/day), while PA-HRB I and PA-HRB II were not different between women and men.

Table 1 presents the characteristics of the study group and comparisons between groups with PA-EE below and above median, calculated separately for women and men. Subjects with PA-EE above median (more active subjects) had higher indices of PA (PA-HRB I, PA-HRB II), and lower homocysteine levels. Other variables were not different.

Table 2 shows TAS, TOS and oxidative markers of the study group and comparisons between groups with PA-EE below and above median. Platelet lipid peroxides were higher in less active subjects. Other variables were not different.

Table 3 contains the information about the main cardiovascular diseases and the medication intake profile. There were no differences between more and less physically active subjects.

Correlations of PA indices with TAS and TOS markers and products of oxidative stress are presented in Table 4. In the whole study population, PA-EE showed negative correlation with platelet lipid peroxides, free thiol and amino groups in platelet proteins, and the concentration of superoxide anion generated by non-homocysteinylated and stimulated platelets. In the group of women, we observed the same correlations except for the PA-EE and free amino groups of platelet proteins. In men, PA-HRBI was positively associated with TAS.

The results of TAS, TOS and oxidative markers were further compared between subjects with at least one PA-HRB II and without it. In the whole population, subjects with at least one PA-HRB II expressed significantly lower TOS (U _Mann–Whitney_ = −2.97; *p* = 0.01) and lower free amino groups of plasma protein (U _Mann–Whitney_ = −2.27; *p* = 0.01). By division of sex, the only association still present was between TOS and PAHRB II (U _Mann–Whitney_ = −3.37; *p* = 0.01) in men.

In the next step, the influence of PA on oxidative stress markers was analyzed in multivariate analyses together with age, sex, BMI, WHR, WHtR, the incidence of cardiovascular disease, medication intake and other cardiovascular risk factors. Independent variables with *p* < 0.05 in univariate analyses were entered into the general linear model. A significant positive impact of PA-HRB II was revealed for TOS. PA-EE expressed positive impact on lipid peroxides and superoxide anion but negative for free thiol and free amino groups in platelets proteins. Figure 1 presents most important associations between PA and redox markers.

In the multivariate model, TOS was influenced by BMI, PA-HRBII, sex and the use of ARB:TOS [mM] = 0.05 + 0.009 × BMI [kg/m^2^] − 0.05 if at least one PA-HRB II + 0.026 if woman − 0.08 if ARB

Platelet lipid peroxides were influenced by vWF and PA-EE:Lipid peroxides _platelets_ [nmol/μg of protein] = −7.68 + 8.45 × vWF [µg/mL] − 0.52 × PA-EE [kcal/kg/day]

Free thiol groups of platelet proteins concentration were influenced by vWF, PA-EE, uric acid and diagnosed lipid disorders:Free thiol groups _platelet proteins_ [μmol/μg of protein] = 99.57 + 8.65 × vWF [µg/mL] − 8.39 × uric acid [mg/dl] − 1.47 × PA-EE [kcal/kg/day] − 12.58 if lipid disorders

The multivariable model selected PA-EE and antiplatelets drugs as independent factors affecting the concentration of free amino groups of platelet proteins:Free amino groups _platelet proteins_ [nmol/μg of protein] = 10.72 − 0.18 × PA-EE [kcal/kg/day] + 1.46 × if present antiplatelet agent

Superoxide anion concentration generated by non-homocysteinylated platelets was influenced by glucose, PA-EE and antiplatelet drugs:Superoxide anion _in platelets PBS 1×10_^8^_plt/mL dilutant_ = 7.43 + 0.41 × glucose [mg/dL] − 0.97 × PA-EE [kcal/kg/day] − 8.56 × if antiplatelet agent

Similarly, in the multivariate analysis, the superoxide anion generated by homocysteinylated platelets was determined by glucose, PA-EE and the use of antiplatelet drugs:Superoxide anion _in platelets Hcy 1×10_^8^_plt/mL dilutant_ = 8.19 + 0.46 × glucose [mg/dL] − 1.10 × PA-EE [kcal/kg/day] − 9.81 × if antiplatelet agent

## 4. Discussion

It seems that this is the first study that assessed the relationship between different aspects of PA and such a large panel of markers of oxidative status in adults entering older age. PA-EE refers to the total energy expenditure spent on different levels of activity (light, moderate, heavy and very heavy during a representative week, related to both work and non-work activity). On the other hand, the Stanford questionnaire assesses the health-related behaviors associated with PA. It is linked with greater awareness of the importance of PA and is often correlated with respondents’ education [39]. Obtained data show that independently of PA assessment method, the impact of PA may be different on oxidative stress markers in platelets as compared to plasma proteins and also dissimilar on platelet lipids and proteins.

For TAS, the only significant association found was the positive correlation between TAS and PA-HRBI in men, and exclusively in bivariate analysis. There are results in the literature showing the relationship between PA and antioxidant potential [6,54,55,56,57]. Occasionally performed physical effort may have a positive effect on the total antioxidant potential [54,55], similar to regular PA [56,57,58]. On the other hand, there are studies showing the negative impact of PA on TAS [59,60]. The discrepancy in the presented results may be related to different methods of assessing PA and its impact on the antioxidant potential.

In the present study, lower TOS was linked to higher PA associated with intensive leisure time exercising (PA-HRBII). This result is consistent with some [54,55,56], but not all [61,62] results of available studies confirming the protective effect of PA on oxidative stress but rather in terms of exercise training or PA connected with intensive health-related behaviors. No studies have been found in the available literature to suggest a relationship between TOS and physical exercise estimated on the basis of energy expenditure.

Plasma lipid peroxides were related to neither PA-EE nor PA-HRB. The observation for platelet lipid peroxides concentration was different. In that case higher vWF and lower PA-EE were associated with higher level of that marker. There are papers indicating positive antioxidative and antiaggregatory effects of PA on platelets; however, for lipid peroxides, it is related to plasma environment [56,57]. Our data suggest that greater daily energy expenditure may play the role of a protective factor against platelet lipids peroxidation and consecutive dysfunction.

Oxidation of protein -SH groups leads to biological malfunction of the proteins, disruption of the cell barrier and free -SH groups loss. In our study, the only parameter which influenced the plasma thiol groups and plasma free amino groups in multivariate design was glucose. None of the PA measures influenced the free thiol groups or free amino groups of plasma proteins. Contrary to plasma, platelet proteins -SH groups were in multivariable analysis influenced negatively by UA, PA-EE and presence of lipid disorders, and positively by vWF. Likewise, consistent negative correlation was observed between platelet free amino groups with PA-EE.

In both non-homocysteinylated and homocysteinylated platelets reaction, the glucose concentration, PA-EE and the use of antiplatelets drugs were selected as determinants of superoxide generation in the multivariate approach. This potential effect of PA confirms previous studies, as PA may induce extracellular superoxide dismutase, an enzyme responsible for attenuating the level of ROS [63]. PA by platelet stress activation down-regulates endothelial angiotensin II type 1 receptor expression. The effect is expressed as decreased NADPH oxidase activity, an superoxide anion production, which finally reduces ROS generation and prolongs endothelial NO bioavailability [64]. The presented results indicate the fact that superoxide generation may be dependent on modifiable factors such as PA level.

The interpretation of the obtained data is certainly not easy. A significant negative correlation between the concentration of lipid peroxides in platelets and PA suggests that PA has a beneficial influence on the structure of platelets and may thus have a health-promoting effect. The platelet compartment particularly rich in lipids is the platelet membrane. Its proper structure is crucial for maintaining the physiological reactivity of platelets [65]. Lipid peroxidation leads to structural changes (i.e., increased rigidity or increased fluidity) of the membranes of various cells [66,67]. In the case of blood platelets, such lipid peroxidation-dependent modulation of membrane physical state may lead to pathological changes in platelet reactivity [68,69].

If we look at our data regarding the relationship between PA and the concentration of free thiol and amino groups in platelet proteins, we may conclude that PA can be perceived as a negative factor, with a pro-oxidant effect, because higher PA is associated with a decrease in the concentration of free thiol and amino groups. Therefore, in this regard PA is similar to different oxidants and activators of blood platelets [70,71]. Such a relationship suggests an increased oxidative stress exerted by PA on platelet proteins. Taking into account the importance of free thiol groups for the preservation of the structure and function of blood platelets [72], an increased oxidation of SH groups may be regarded as a risk factor for unwanted changes in platelet physiology.

In our study, we did not attempt to determine the source of ROS appearing in platelets and blood plasma as a result of PA. However, we know that the greater the PA, the less generation of superoxide anion radical by control and homocysteinylated platelets. Unambiguous interpretation of these results is difficult. Lower concentration of superoxide anion generated by platelets under control conditions in subjects showing higher PA may explain the lower peroxidation of platelet lipids. However, it contradicts the increased degree of oxidation of thiol and amino groups of platelet proteins. Usually, generation of superoxide anion is paralleled by oxidation of thiol or amino protein groups. Perhaps, in our case, the superoxide anion interacts mainly with membrane lipids. Increased PA may lead to less superoxide anion radical generation and less frequent attacks on membrane lipids. The same applies to superoxide anion generation by homocysteinylated platelets, which decreases with increasing PA. At first glance, this observation supports the antioxidant (protective, prophylactic) effect of PA, since higher level of superoxide anion can be found after homocysteinylation of blood platelets [70]. It should be remembered, however, that the homocysteinylation reaction takes place primarily through the binding of homocysteine to free thiol and amino groups of proteins [73]. The observed reduced superoxide anion generation during in vitro homocysteinylation suggests that blood platelets (especially proteins of platelet membrane) may have fewer anchor sites able to react with homocysteine, i.e., reduced concentration free thiol and amino groups of the proteins. These scenarios need to be verified in future experiments.

In the current article, several markers of oxidative damage of proteins and lipids in platelets and plasma were presented. Among the reactive oxygen species, we paid special attention to the superoxide anion radical. However, it should be remembered that other reactive oxygen species, omitted in our analyses, are important in shaping the physiological and pathological responses of platelets to different stimuli. Hydrogen peroxide, the precursor of which is a superoxide anion radical, is an intraplatelet mediator of signaling pathways related to the stimulation of platelet aggregation by collagen and thrombin [74,75]. It is consistent with an observation that the degree of oxidative damage to platelet lipids is directly related to the activity of antioxidant enzymes, including hydrogen peroxide-scavenging catalase: the higher the catalase activity, the lower the degree of lipid peroxidation [76]. The role of hydrogen peroxide in shaping platelet reactivity is not simple and unambiguous. As mentioned above, intraplatelet hydrogen peroxide is a mediator of pro-aggregation activity of collagen, but this seems to be true in the case of endogenous, intraplatelet hydrogen peroxide. Hydrogen peroxide acting on platelets as an extra platelet agent, in high but non-toxic concentrations, exerts antiaggregatory effects and inhibits the action of various platelet agonists [77]. Moreover, the effect of hydrogen peroxide on the oxidative damage of different biomolecules (proteins versus lipids) is not the same [78]. In subsequent studies on the importance of ROS and oxidative stress for disorders of the structure of platelets and plasma proteins, these various relationships should be taken into account since they significantly contribute to platelet activation and reactivity [79]. In addition, one should remember the key role of interactions of various ROS and various enzymatic antioxidant systems, especially superoxide dismutase, catalase, glutathione peroxidase, whose activities change under conditions of oxidative stress modulated or not modulated by exogenous antioxidants [78,80].

The variety of interactions between oxidants and antioxidants in platelets can be difficult to grasp. An example of such a relationship may be the protection of the intraplatelet pool of tocopherol against oxidation by ascorbate (this is a beneficial effect considering the antiplatelet effect of tocopherol) or the fact that platelet glutathione peroxidase enhances the antiplatelet effect of nitric oxide. Understanding these relationships in order to avoid simplistic conclusions may be crucial for designing effective antiplatelet therapies based on antioxidants that achieve the desired therapeutic goal and avoid undesirable side effects (e.g., already recorded in men taking vitamin E). The currently available data on such an antiplatelet effect related to the modulation of the redox status of platelets, noted for dipyridamole or red wine polyphenols, seem to be promising [81].

In the light of abovementioned relations and obtained observations, we may describe the PA-dependent oxidative stress as difficult to be unambiguously assessed from the point of view of cardiometabolic risk assessment, when platelets are considered an important point of influence of ROS.

## 5. Conclusions

Independently of PA character, similar advantageous effects of PA were increased TAS, and decreased TOS and specific oxidative platelet lipids stress markers. However, the association of PA was negative to free thiol and free amino groups in platelets proteins. Therefore, the impact of PA may be more visible for platelets than plasma markers. It may also be dissimilar on platelet lipids and proteins: for lipids oxidation, PA seems to have protective effect, while for platelets proteins, PA tends to act as pro-oxidative factor. However, it appears that knowledge in this area is still limited, though this study can become the basis for further research.

## 6. Limitations of the Study

One of the significant limitations of the presented study is its cross-sectional nature and any cause–effects relationship should not be concluded. The presented results concern the population of elderly people from central Poland. In addition, determinations of physical activity are based on a questionnaire study, which may be associated with the effect of inaccurate estimation. Assessment of physical activity with a questionnaire is always subject to some bias. Nevertheless, both questionnaires used in the study have been widely described and validated. The limitation of TAS and TOS methods is the lack of distinction between individual oxidants and antioxidants (low molecular weight molecules, enzymatic proteins and non-enzymatic proteins). There are also no established reference ranges allowing for classification into the group with normal, reduced or elevated TAS/TOS. Finally, both calorie intake and inflammation processes may be significant co-determinants of oxidative and antioxidant potential and future studies should also take into account dietary data and inflammation markers.

## Figures and Tables

**Figure 1 antioxidants-12-01200-f001:**
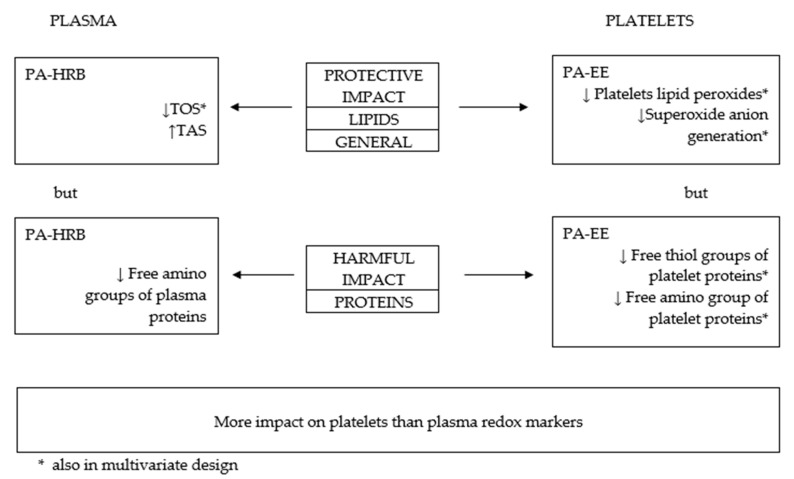
Graphical presentation of the main findings. PA-EE, physical activity–energy expenditure; PA-HRB, physical activity–health-related behavior. ↑, positive relationship; ↓, negative relationship.

**Table 1 antioxidants-12-01200-t001:** Characteristics of the study group and comparisons between group with PA-EE below and above median.

	Subjects with PA-EE <Me N = 150	Subjects with PA-EE ≥Me N = 150
Age [years]	63 (62–64)	63 (61–64)
Education [years]	13 (12–17)	13 (12–15)
BMI [kg/m^2^]	27.71 (25.27–30.55)	27.86 (24.77–31.27)
WHR	0.93 (0.84–1.00)	0.93 (0.85–1.00)
WHtR	0.57 (0.53–0.61)	0.57 (0.52–0.63)
Obesity (BMI ≥ 30 kg/m^2^)	48 (32%)	47 (31%)
PA-EE [kcal/kg/day]	39.14 (36.53–40.75)	50.41 (46.71–54.85) ***
PA-HRB I	3.0 (1.0–4.0)	3.0 (2.0–4.0) *
PA-HRB II (at least 1); n (%)	12 (8%)	35 (23.3%) *
SBP [mmHg]	135 (123–145)	136.5 (127–150)
DBP [mmHg]	82 (75–93)	84 (75–92)
Pulse[/min]	67 (62–72)	66.5 (62–72)
Blood platelets [10^3^/mm^3^]	207.5 (180.0–239.5)	214 (181–252)
Total cholesterol [mg/dL]	202.15 (170.2–231.2)	205.6 (172.5–244.5)
LDL cholesterol [mg/dL]	125.6 (100.8–151.1)	126.5 (100.7–160.7)
HDL cholesterol [mg/dL]	48.3 (40.2–57.1)	48.05 (41.4–58.4)
Triglycerides [mg/dL]	111.7 (82.6–155.8)	110.1 (72.6–165.6)
Glucose [mg/dL]	99.2 (90.7–108.3)	100.6 (92.6–113.5)
Uric acid [mg/dL]	5.0 (4.1–5.8)	4.7(4.0–5.6)
Homocysteine [µmol/L]	15.0 (13.2–17.3)	14.0 (12.1–16.8) *
vWF [µg/mL]	5.54 (4.91–6.09)	5.45 (4.84–6.21)
VCAM [ng/mL]	269.0 (246.5–297.0)	274.2 (253.4–300.2)
ICAM [ng/mL]	209.0 (202.9–218.3)	211.7 (202.9–219.1)
TXB2s [pg/mL]	8.41 (7.44–8.88)	8.18 (7.42–8.91)
TXB2d [pg/mL]	9.03 (8.12–9.58)	8.87 (7.9–9.64)

* *p* < 0.05; *** *p* < 0.001; significantly different in comparison with the group with PA-EE below median (Me). Data presented as median (lower-upper quartile). Comparisons between group with PA-EE below and above median were performed with the use of the Mann–Whitney U test, the χ2 test or the Fisher exact χ2 test. Abbreviations: BMI: body mass index; WHR: waist–hip ratio; WHtR: waist–height ratio; PA-EE: physical activity–energy expenditure; PA-HRBI: physical activity-moderate health related behaviors; PA-HRBII: physical activity-intensive health-related behaviors; SBP: systolic blood pressure; DBP: diastolic blood pressure; LDL-cholesterol: low-density lipoprotein cholesterol, HDL-cholesterol: high-density lipoprotein cholesterol; vWF: von Willebrand factor; VCAM-1: vascular cell adhesion protein 1; ICAM-1: intracellular adhesion molecule 1; TXB2s- thromboxane 2 generated in static conditions; TBX2d thromboxane 2 generated in dynamic conditions; TAS: total antioxidant status; TOS: total oxidative status.

**Table 2 antioxidants-12-01200-t002:** TAS, TOS and oxidative markers of the study group and comparisons between group with PA-EE below and above median.

	Subjects with PA-EE <Me N = 150	Subjects with PA-EE ≥Me N = 150
TAS [mM]	42.19 (31.07–46.85)	40.99 (30.36–46.37)
TOS [mM]	0.53 (0.08–0.59)	0.54 (0.08–0.60)
Plasma lipid peroxides [mmol/L]	0.29 (0.01–0.84)	0.26 (0.02–1.64)
Platelet lipid peroxides [nmol/μg of protein]	1.32 (0.61–25.95)	0.94 (0.41–3.02) *
Free thiol groups of platelet protein [μmol/μg of protein]	2.89 (1.96–44.2)	2.71 (1.83–4.69)
Free thiol groups of plasma protein [μmol/μg of protein]	0.03 (0.02–0.04)	0.03 (0.02–0.04)
Free amino groups of platelet protein [nmol/μg of protein]	0.19 (0.05–2.20)	0.94 (0.41–3.02)
Free amino groups of plasma protein [mmol/mg of protein]	17.85 (12.21–25.60)	15.82 (9.04–26.35)
Superoxide anion generated by resting platelets [1 × 10^8^ plt/mL dilutant]	0.32 (0.12–4.78)	0.41 (0.12–1.72)
Superoxide anion generated by homocysteinylated platelets [1 × 10^8^ plt/mL dilutant]	0.38 (0.14–7.09)	0.45 (0.15–2.11)

* *p* < 0.05; * significantly different in comparison with the group with PA-EE below median (Me). Data presented as median (lower -upper quartile). Comparisons between group with PA-EE below and above median were performed with the use of the Mann–Whitney U test. Abbreviations: TAS -total antioxidant status; TOS- total oxidative status.

**Table 3 antioxidants-12-01200-t003:** Medical records and medication profile—comparisons between group with PA-EE below and above median.

	Subjects with PA-EE <Me N = 150	Subjects with PA-EE ≥Me N = 150
Metabolic syndrome	93	102
Arterial hypertension	82	75
Hypercholesterolemia	100	98
Diabetes mellitus type 2	17	18
Myocardial infarction	11	4
Chronic ischemic heart disease	17	26
Previous stroke	5	7
Smoking	33	36
Antiplatelets drugs	30	24
β-adrenolytic drugs	43	42
Ca-blockers	16	17
Angiotensin converting enzyme Inhibitors	37	33
Angiotensin II receptor blockers	15	12
Diuretics	25	32
Hypolipidemic drugs	33	36
Antidiabetic drugs	15	19

Comparisons between group with PA-EE below and above median (Me) were performed with the use of the χ2 test or the Fisher χ2 test.

**Table 4 antioxidants-12-01200-t004:** Correlations of PA-EE and PA-HRBI with TAS, TOS and oxidative stress markers.

		PA Indices	TAS	TOS	Platelet Lipid Peroxides	Plasma Lipid Peroxides	Free Thiol Groups of Platelet Protein	Free Thiol Groups of Plasma Protein	Free Amino Groups of Platelet Protein	Free Amino Groups of Plasma Protein	Superoxide Anion Generated by Non-Homocysteinylated Platelets	Superoxide Anion Generated by Homocysteinylated Platelets
Correlations	Whole study population	PA-EE [kcal/kg^−1^ day^−1^]	−0.07	−0.04	−0.12 *	0.08	−0.15 *	−0.01	−0.15 *	−0.12	−0.14 *	−0.14 *
		PA-HRB I	0.09	−0.01	0.08	0.11	0.02	−0.03	0.07	0.02	−0.02	−0.01
	Women	PA-EE [kcal/kg^−1^ day^−1^]	−0.13	−0.07	−0.18 *	0.15	−0.28 *	−0.05	−0.16	−0.15	−0.21 *	−0.21 *
		PA-HRB I	0.01	−0.05	0.06	0.12	0.07	−0.03	0.04	0.01	−0.05	−0.05
	Men	PA-EE [kcal/kg^−1^ day^−1^]	−0.10	−0.03	−0.08	−0.02	−0.07	0.09	−0.14	−0.04	−0.07	−0.07
		PA-HRB I	0.18 *	0.02	0.09	0.10	0.00	0.01	0.10	0.04	0.02	0.03

* *p* value < 0.05. Correlations were performed with Spearman’s rank correlation test (rho coefficient).

## Data Availability

The raw data used to support presented findings may be obtained by sending a request to the corresponding author.

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
