# Peer review of "Contribution of Physical Activity to the Oxidative and Antioxidant Potential in 60–65-Year-Old Seniors"

_antioxidants, 2023, doi:10.3390/antiox12061200_

Round 1

Reviewer 1 Report (Previous Reviewer 4)

The manuscript  has significantly improved after revision, although it would have been useful to insert graphics, to help the reader.

best

no comment for English language

Author Response

Reviewer 1

The manuscript  has significantly improved after revision, although it would have been useful to insert graphics, to help the reader.

Author response: According to the suggestions of all the four reviewers, we changed the presentation of the data. We tried to elaborate  figures that would enable better presentation of the results. Nevertheless, it was difficult to present clearly the results on the graph of several multiple regression analyses, each with usually several independent selected predictors. However, we have prepared a graphic with the most important results of our research.

Reviewer 2 Report (Previous Reviewer 3)

The authors of the manuscript entitled “Contribution of physical activity to the oxidative and antioxidant potential in 60-65-year-old seniors“ answered all the previous  comments and nicely completed the manuscript. Nevertheless, corrections to minor errors and text editing are still needed.

In Material and Methods, the authors added part 2.2 Physical activity. Nevertheless, line 96-104 introducing the Seven-Day Recall Physical Activity Questionnaire and the Stanford Usual Activity Questionnaire should be placed in part 2.2

Line 88: 300 should be written in letters.

Line 184-185: the space between line 184-185 should be modified.

Line 218-220: the sentence “More active subjects had higher all indices of PA, and lower homocysteine levels.” is not clear and should be rewritten. For example,“Subjects with PA-EE above median (more active subjects) had higher indices of PA (PA-HRB I ,PA-HRB II), and lower homocysteine levels.”

Author Response

Reviewer 2

The authors of the manuscript entitled “Contribution of physical activity to the oxidative and antioxidant potential in 60-65-year-old seniors“ answered all the previous  comments and nicely completed the manuscript. Nevertheless, corrections to minor errors and text editing are still needed.

In Material and Methods, the authors added part 2.2 Physical activity. Nevertheless, line 96-104 introducing the Seven-Day Recall Physical Activity Questionnaire and the Stanford Usual Activity Questionnaire should be placed in part 2.2

Author response: This part of Material and Methods was incorporated into subparagraph 2.1. as it was  suggested.

Line 88: 300 should be written in letters.

Author response:  The number of participants was placed in letters, as it was suggested.

Line 184-185: the space between line 184-185 should be modified.

 Author response: Unnecessary space was removed.

Line 218-220: the sentence “More active subjects had higher all indices of PA, and lower homocysteine levels.” is not clear and should be rewritten. For example, “Subjects with PA-EE above median (more active subjects) had higher indices of PA (PA-HRB I ,PA-HRB II), and lower homocysteine levels.”

Author response: this sentence was rewritten in accordance to the Reviewer’s suggestion.

Reviewer 3 Report (Previous Reviewer 2)

The manuscript is interesting and aims to relate the level of physical exercise with the antioxidant potential in 60-65 year old subjects.

Some aspects need to be cleared before the manuscript can be accepted for publication.

First of all, I ask the authors to justify in the manuscript the choice of selecting a sample of 60-65 years old. Why not 65-70? or 70-75?

In any case, a 5-year window could be limiting.

I agree with the authors that some aspects of the assessment of redox status/antoxidant status/oxidant status may vary greatly between plasma and cell tests. The study focuses heavily on platelets. However, this is a limitation if a generalized study is placed in the title.

So why not examine cells particularly sensitive to oxidative stress such as erythrocytes? What about other cell types like lymphocytes?

Isn't it better to rethink the manuscript completely on the influence of (estimated) PA on platelets? Perhaps it is better to focus on just one aspect given all the limitations set out in paragraph 6.

Please some sentences need linguistic proofreading.

Author Response

Reviewer 3

The manuscript is interesting and aims to relate the level of physical exercise with the antioxidant potential in 60-65 year old subjects.

Some aspects need to be cleared before the manuscript can be accepted for publication.

First of all, I ask the authors to justify in the manuscript the choice of selecting a sample of 60-65 years old. Why not 65-70? or 70-75?

In any case, a 5-year window could be limiting.

Author response: The presented research material is a fragment of a broader research context. The choice of such an age window was not accidental. We wanted to find respondents on the borderline between still professionally active ones and those who are  retired. Secondly, the population of young seniors is extremely interesting because, on the one hand, it often retains activity as in the middle age, on the other hand, markers related to the aging of the cardiovascular system are already present. Thirdly, the age limit and gender balance were to provide the homogeneity of the study group. All these aspects have been outlined in the introduction and  discussion.

I agree with the authors that some aspects of the assessment of redox status/antoxidant status/oxidant status may vary greatly between plasma and cell tests. The study focuses heavily on platelets. However, this is a limitation if a generalized study is placed in the title.

Author response: As a matter of fact, the study focuses heavily on platelets. Nevertheless, the study addresses the redox state in the two parallel environments. Blood platelets, as noted by the Reviewer, but also plasma. It is plasma that is the buffer for total oxidative and antioxidant capacity. Hence, looking at two environments in our study, we attempted to generalize the relationship between oxidation and physical activity in the title.

So why not examine cells particularly sensitive to oxidative stress such as erythrocytes? What about other cell types like lymphocytes?

Author response: We are fully aware of some limitations of our study. Hypothetically, we can assume that by examining the oxidative stress in various blood cells, we could obtain various results at the same time. What's more, the inclusion of plasma macrostructures (both protein and lipid) in the study would give an even broader insight into the discussed issue. However, like any study, ours was also determined by the financial and technical framework, hence we had to choose the laboratory determinations that were significant in our primary assessment. The need of further assessment in different blood cells has been acknowledged in the final part of the discussion.

Isn't it better to rethink the manuscript completely on the influence of (estimated) PA on platelets? Perhaps it is better to focus on just one aspect given all the limitations set out in paragraph 6.

Author response: As stated above, we concentrated heavily on platelets, but the concomitant assessment of TAS, TOS and plasma lipids and proteins gives more comprehensive insight into the complex assessment of the potential relationship of physical activity to redox state. As mentioned earlier, future inclusion of different blood cells should also be considered.

Comments on the Quality of English Language

Please some sentences need linguistic proofreading.

Author response: The manuscript was amended by the Native Speaker as  suggested.

Reviewer 4 Report (Previous Reviewer 1)

The authors have presented the results obtained that were forgotten in the first version of this manuscript. The changes introduced allow to understand its study and to make a review.

They estimate how the physical activity indirectly determined may relate to the total antioxidant status, total oxidative status and the variety of plasma and platelet markers of an oxidation in a population of 300 younger seniors. The study has an interest evidencing different oxidative damage patterns in plasma and platelets in response to physical activity.

However, it is adequate to consider some minor questions in relation to the presentation and redaction of this paper.

Abstract:

Lines 9-10: Avoid indicate that ‘there are no publications differentiating the relationships between different targets’ because there are publications differentiating the relationships between different targets. See the references PMID: 32178436. PMID: 30453505. PMID: 30347790. PMID: 16481153. PMID: 32230858. PMID: 16595757

Lines 21-23: Indicate the kind of impact: it was positive or it was negative for the plasma TOS level? Similarity indicates the direction of the impact for all parameters referred to the platelets. It would be better if you indicate the parameters of platelets that were affected by the PA. The concept in the abstract of ‘All the monitored’ markers is very ambiguous.

Line 26: What do ’protective effect contrary’ mean? What does this phrase mean?

Introduction

Lines 67-70: There are studies with results on the effects of physical activity on the production of the superoxide anion and oxidative stress in older athletes and also in the older population. (see PMID: 27735833).

Results:

Lines 248-254: The correlation values indicate a very poor correlation, although statistically significative, between these parameters with values below -0,28. You can use this concept in your discussion.

Lines 264-266: Indicate the kind of impact: it was positive or it was negative for the plasma TOS level? Similarity indicates the direction of the impact for all parameters referred to the platelets. It would be better if you indicate the parameters of platelets that were affected by the PA. The significant impact on TOS plasma was for PA-HRB or for PA-HRBII?

Discussion

Lines 291-292 Avoid affirming something without evidence. In the literature you can find studies that assessed the relationship between different aspects of PA and such a large panel of markers of oxidative status in adults entering older age.

Lines 306-307: What discrepancy? Your results discrepant of those showing a negative or of those a positive impact of PA on TAS

Line 309: lower TOS… in plasma?

Lines 322-332: What do you discuss in this paragraph? What does the reference 56 in the final of the phrase mean? Perhaps your results indicate that increasing the total amount of FA contributes to decrease the free thiol groups and free amino groups in platelet proteins and, then it ‘leads for biological inactivation of the protein, oxidation of membrane thiol groups and disintegration of the cell barrier’ as indicate in this paragraph?

Lines 333-337:  if PA may induce extracellular superoxide dismutase, an enzyme responsible for attenuating the level of ROS, then why PA-EE  is a decreasing determinant of superoxide generation by platelets? Your ‘in vitro’ extracellular measuring superoxide anion production by platelets does not allow to induce the production of extracellular superoxide dismutase. It seems that platelets have the lower capability to produce superoxide anion, not that platelets have higher capabilities to eliminate superoxide anion.

Lines 343-351: What do you discuss?

Lines 361-373: it is very speculative.

Lines 364-367: The generation of superoxide anion is extracellular as you indicate, but the damaged proteins are cellular. It is difficult to intent explain

Lines 374-382: It is very speculative.

Lines 383-406: This discussion is not related with your results and not contribute to attain the conclusion.

Lines407-416: It could be interesting in other context differences of this study.

Lines 417-420: Is the PA-dependent oxidative stress difficult to be unambiguously assessed from the point of view of cardiometabolic risk assessment?

Lines 419-420: Are the platelets considered as the main point of the influence of ROS production? Your study does not consider the degree of platelets’ contribution to the vascular oxidative stress status. You observe that there are influences on the oxidative stress status of platelets that do not have relevance on the oxidative stress stratus of plasma.

Lines 422: Is the first study to compare the two aspects of with a wide range of oxidative stress markers, concomitantly, in both plasma and platelet proteins? It is a conclusion?

Lines 425-430: It doesn't reflect the main results obtained.

Author Response

Reviewer 4.

The authors have presented the results obtained that were forgotten in the first version of this manuscript. The changes introduced allow to understand its study and to make a review.

They estimate how the physical activity indirectly determined may relate to the total antioxidant status, total oxidative status and the variety of plasma and platelet markers of an oxidation in a population of 300 younger seniors. The study has an interest evidencing different oxidative damage patterns in plasma and platelets in response to physical activity.

However, it is adequate to consider some minor questions in relation to the presentation and redaction of this paper.

Abstract:

Lines 9-10: Avoid indicate that ‘there are no publications differentiating the relationships between different targets’ because there are publications differentiating the relationships between different targets. See the references PMID: 32178436. PMID: 30453505. PMID: 30347790. PMID: 16481153. PMID: 32230858. PMID: 16595757

Author response:  The above sentence has been edited to indicate more precisely the area of the study - elements of plasma and platelets that regulate oxidative stress. Suggested references have been included to the manuscript.

Lines 21-23: Indicate the kind of impact: it was positive or it was negative for the plasma TOS level? Similarity indicates the direction of the impact for all parameters referred to the platelets. It would be better if you indicate the parameters of platelets that were affected by the PA. The concept in the abstract of ‘All the monitored’ markers is very ambiguous.

Author response: We specified in the abstract that we obtained a positive effect of PA-HRB on TOS. As for the term "all the monitored markers of oxidative stress measured in platelets proteins", the sentence has been reformulated to indicate clearly the obtained associations.

Line 26: What do ’protective effect contrary’ mean? What does this phrase mean?

Author response: This phrase refers to different, bidirectional association between PA and platelets structures. In terms of lipids PA seems to have protective effect, acts positively. Concomitantly, for platelets proteins (studied thiol and amino rests) PA seems to have more deteriorating effect. Yet, we have rewritten that part to make it clearer.

Introduction

Lines 67-70: There are studies with results on the effects of physical activity on the production of the superoxide anion and oxidative stress in older athletes and also in the older population. (see PMID: 27735833).

Author response: The suggested study was testing 5 younger and 5 older subjects. In addition, the work is very interesting and extensive in terms of oxidative stress factors analysed from physical activity, but it does not examine specific markers of protein damage, such as amino and thiol rests. However, the work has been added to the citation.

Results:

Lines 248-254: The correlation values indicate a very poor correlation, although statistically significative, between these parameters with values below -0,28. You can use this concept in your discussion.

Author response: The number of participants in the present study is higher than in the majority of previous reports on redox state and physical activity. Therefore, we could detect even lower correlations that were statistically significant.

Lines 264-266: Indicate the kind of impact: it was positive or it was negative for the plasma TOS level? Similarity indicates the direction of the impact for all parameters referred to the platelets. It would be better if you indicate the parameters of platelets that were affected by the PA. The significant impact on TOS plasma was for PA-HRB or for PA-HRBII?

Author response: The mentioned association between TOS and PA was positive and involved PA-HRBII (subjects with at least one PA-HRBII had lower TOS). In addition, we have added detailed information about the relationship between PA and platelet parameters as suggested.

Discussion

Lines 291-292 Avoid affirming something without evidence. In the literature you can find studies that assessed the relationship between different aspects of PA and such a large panel of markers of oxidative status in adults entering older age.

Author response: We searched the literature and we didn’t find studies that assessed the relationship between different aspects of PA and such a large panel of markers of oxidative status in adults entering older age.

Lines 306-307: What discrepancy? Your results discrepant of those showing a negative or of those a positive impact of PA on TAS

Author response: Discrepancy concerns different relationship of PA to redox parameters in previous studies, as cited in the two preceding sentences.

In our study, discrepancy concerns different relationship of PA indices to plasma and platelet redox state, and different association of PA to platelet lipids and proteins. This fragment has been modified to present those data more clearly.

Line 309: lower TOS… in plasma?

Author response: Total antioxidant status (TAS) and total oxidant status (TOS) were evaluated in collected samples of blood plasma.

Lines 322-332: What do you discuss in this paragraph? What does the reference 56 in the final of the phrase mean? Perhaps your results indicate that increasing the total amount of FA contributes to decrease the free thiol groups and free amino groups in platelet proteins and, then it ‘leads for biological inactivation of the protein, oxidation of membrane thiol groups and disintegration of the cell barrier’ as indicate in this paragraph?

Author response: As a matter of fact, the results indicate that increasing the total amount of PA contributes to decrease the free thiol groups and free amino groups in platelet proteins. This may lead to biological inactivation of the protein, oxidation of membrane thiol groups and disintegration of the cell barrier. We have modified this paragraph for better presentation.

Lines 333-337:  if PA may induce extracellular superoxide dismutase, an enzyme responsible for attenuating the level of ROS, then why PA-EE  is a decreasing determinant of superoxide generation by platelets? Your ‘in vitro’ extracellular measuring superoxide anion production by platelets does not allow to induce the production of extracellular superoxide dismutase. It seems that platelets have the lower capability to produce superoxide anion, not that platelets have higher capabilities to eliminate superoxide anion.

Author response: We have modified this paragraph. Here again, it seems that the relationship of PA to superoxide generation may be different for plasma and platelets.

Lines 343-351: What do you discuss?

Author response: This paragraph discusses the association between lipid peroxides in platelets and PA.

Lines 361-373: it is very speculative.

Author response: It is a bit, but we feel that different scenarios should be presented. We also clearly state that “this scenario needs to be verified in future experiments”.

Lines 364-367: The generation of superoxide anion is extracellular as you indicate, but the damaged proteins are cellular. It is difficult to intent explain

Author response: In this paragraph we discuss the generation of superoxide anion radical by control and homocysteinylated platelets. We have modified this paragraph to improve the clarity of the presentation.

Lines 374-382: It is very speculative.

Author response: Here again, we try to highlight those complex and not easily explainable mechanisms of potential impact of PA on redox state in plasma and platelets. We have modified this paragraph to improve the clarity of the presentation.

Lines 383-406: This discussion is not related with your results and not contribute to attain the conclusion.

Author response: We believe that we discuss here, in a wider aspect, the potential mechanisms of modulating the redox state based on the results obtained in the present study. Such an approach should enable to present conclusions taking into account the limitations and gaps of current knowledge. It also takes into account the recommendations of other reviewers.

Lines407-416: It could be interesting in other context differences of this study.

Author response: We believe that we tried to highlight here wider aspects of PA-redox relationships, also in the context of the results as the differences found in the present study.

Lines 417-420: Is the PA-dependent oxidative stress difficult to be unambiguously assessed from the point of view of cardiometabolic risk assessment?

Author response: Yes, as some associations seem to present protective while other detrimental impact on the oxidative stress markers.

Lines 419-420: Are the platelets considered as the main point of the influence of ROS production? Your study does not consider the degree of platelets’ contribution to the vascular oxidative stress status. You observe that there are influences on the oxidative stress status of platelets that do not have relevance on the oxidative stress stratus of plasma.

Author response: We agree, this phrase has been modified to  highlight those aspects better.

Lines 422: Is the first study to compare the two aspects of with a wide range of oxidative stress markers, concomitantly, in both plasma and platelet proteins? It is a conclusion?

Author response: We have modified this paragraph.

Lines 425-430: It doesn't reflect the main results obtained.

Author response: We have modified this paragraph to show the main findings and conclusions better.

Round 2

Reviewer 4 Report (Previous Reviewer 1)

The last version of the manuscript corrects all questions and the explanations reported by the authors about maintain the paragraphs in the discussion are convinced. The new redaction of the conclusion is more clear, and it is in accordance with the main results obtained.

This manuscript is a resubmission of an earlier submission. The following is a list of the peer review reports and author responses from that submission.

Round 1

Reviewer 1 Report

They estimate how the physical activity indirectly determined may relate to the total antioxidant status, total oxidative status and the variety of plasma and platelet markers of an oxidation in a population of 300 younger seniors. The study has an interest evidencing different oxidative damage patterns in plasma and platelets in response to physical activity. However, there are major deficiencies in the writing of the Materials and Methods and in the presentation of the results of this study that make it impossible to know how it was carried out and what results were obtained. On the other hand, Are the determinations of such a large number of variables necessary to carry out the study and achieve the proposed objectives?

Materials & Methods

Lines 83-84: Indicate the approval references of this study.

Lines 102-118. The description of the “2.1. Laboratory measurements and 2.2. Estimation of TAS, TOS and markers of oxidative damage of proteins and lipids in blood platelets and plasma” make it difficult in the manuscript to follow the procedures used for these analyses. It will be better to make a brief description of each determination citing the original methods for their determinations. What are the bases of the method to determine the production of superoxide anion by platelets? Why determine in the presence or absence of homocysteine? Is this method specific for the determination of superoxide anion or does it interfere with other reductants of cytochrome c? How were free amino groups and free thiol groups determined in proteins? Among other issues that allow a good description of the methods.

Lines 119-141. Indicates what statistical analysis was used by each result obtained.

RESULTS

No results are showing. The results show the statistical analysis performed of results not shown. It is necessary shown the results obtained and include the information of the statistical analysis performed.

Were the Spearman's rank correlation test or the Pearson's correlation test used to analyse the results presented in the Table 2? What are the indices shown in the Table 2?

What kind of statistical analysis was performed on the results presented in the Table 3?

What is the meaning of parameters such as Free amino groups of proteins or Free thiol groups of proteins? In general, what was the results obtained in this study? You did not present the results obtained in yours analysis and used to perform the statistical analysis.

You describe in Material & Methods the determination of a lot of parameters in plasma and platelets (Lines 103-107) such as glucose, cholesterol, triglycerides, uric acid, homocysteine, VCAM1, ICAM-1, thromboxane B2, etc., but no results are presented nor statistical analysis was performed with these parameters.

Line 218: “Subjects taking antiplatelets drugs….” What kind of subjects are? No mention of this condition was made in Material& Methods nor in the purpose of the study.

Discussion

The discussion seems more an introduction than a discussion of the results. Some aspects of the discussion are better as an introduction of the manuscript.

Lines 235-237:  A statistical comparison between plasma and platelet parameters did have been performed and presented in the results? The results obtained in plasma have not been compared with those obtained in platelets, at least those that are comparable.

Reviewer 2 Report

The manuscript antioxidants-2193245 is interesting and intends to relate physical activity with the oxidant and antioxidant potential in plasma and platelets of a population of adults aged 60-65 years.

The manuscript would have potential but some critical issues do not allow the manuscript to be published in this version.

Here are only some considerations:

First, physical activity is self-reported. Can't it be quantified in some way? If oxidative stress / antioxidant-oxidant capacity / level of physical activity is evaluated, perhaps this point should be clarified and expanded.

The study is substantially an extension of the previous works: doi: 10.1136/bmjopen-2018-025905 and doi: 10.3390/antiox11061065.

Much of the data is reported in these earlier publications. Since the anthropometric data and the values of the parameters relating to oxidative stress are not present in the manuscript, the study definitely lacks power.

Reference is often made in the comments to already published data, compromising the novelty of the work which should instead be highlighted.

It would have been interesting given that it was decided to evaluate plasma and platelets, also examine erythrocytes, specialized cells particularly susceptible to oxidative stress.

Reviewer 3 Report

The manuscript entitled “Contribution of physical activity to the oxidative and antioxidant potential in 60-65-year-old seniors “ studied the impact of physical activity on oxidative stress markers in platelets and plasma proteins. Introduction and Material and Methods are well described. Discussion should be completed. Limitations of the techniques should be added.

Physical activity assessment should be briefly added. Furthermore, the authors should mention the limitations of the questionnaire because there is no single best method that can assess all aspects of physical activity and energy expenditure.

The authors used TAS and TOS methods. Could they present the limitation of both techniques?

The authors only studied superoxide dismutase. Studies in Human have also demonstrated that platelet activation is associated with production of H2O2 and pre-treatment with catalase eliminate production of H2O2. Therefore, the authors should discuss the importance of catalase.

As age dependant platelet hyper-activation is mediated by increase platelet O2 generation ant to its conversion to H2O2 intracellularly. Could the authors discuss the importance of both glutathione peroxidase and the superoxide dismutase and of their synchronous antioxidant protection? Furthermore, in platelet GSH depletion attenuates GPX activity and induce an increase in lipid peroxidation altering redox homeostasis (Friedman et al 2008 Arter. Thromb. Biol). However, H2O2 decomposition could be also catalysed by the thiol-selenoperoxidase peroxiredoxins (jang et al; j boil Chem 2015).

Line 238: the authors mentioned “For TAS, the only significant association found was the positive correlation between TAS and PA-HRBI in men” and Line 252, the authors mentioned “In that case higher vWF and lower PA-EE were associated with higher level of Plasma lipid peroxides”. Could the authors show the level of TAS in the case of higher vWF.

Furthermore, the conclusion line 255-257 is too strong line 255 “Therefore, our study seems to be the first to indicate that mechanism platelet’s peroxidation and consecutive dysfunction”. The authors should modify the sentence. Furthermore, line 288-291 is redundant with line 255-257.

It is not because lower PA-EE is correlated to higher level of plasma lipid peroxides that the inverse is true. This affirmation should be modulated by the authors. The authors should also consider the calorie intakes. High physical activity and EE is in balance with the calorie intake that might be lower in the population of seniors. This point should be added in the discussion.

Line 267 the authors wrote “The fact that platelet free amino groups, as previously free thiol groups, were significantly lower among those with higher PA-EE, may once more indicate 268 on the specific effect of PA on platelet redox balance[41].” The sentence should be modulated because no correlation is observed and it is only speculation.

The authors could consider inflammation and reactive oxygen species because some markers of inflammation are intricately linked to reactive oxygen species. Physical activity levels and inflammation should be discussed. Physical activity improves antioxidant defences and lower lipid peroxidation levels in aged individuals. Elderly physically active individuals show antioxidant activity and lipid peroxidation levels similar to young sedentary subjects, emphasing the importance of physical activity in aging (Bouzid et al int J sport med 2018). This reference should be added.

Reviewer 4 Report

In this article, the authors study 300 participants from central Poland, regarding the acute exercise and regular physical activity (PA) measuring Total antioxidant potential (TAS), total oxidative stress (TOS) and several other markers of an oxidative stress, monitored in platelet and plasma proteins. The authors concluded that independently of PA assessment method, the impact of PA may be different on oxidative stress markers in platelets as compared to plasma proteins and also dissimilar on platelet lipids and proteins. These associations are more visible for platelets than plasma markers. For lipids oxidation, PA seems to have protective effect contrary to platelets proteins, where PA tends to act as pro-oxidative factor.

1.Introduction:

This study is a part of a bigger one carried out on “The occurrence of oxidative stress and selected factors for cardiovascular risk and functional status of older people in the context of workload”. Two scientific papers have been already published SoÅ‚tysik and al. Antioxidants 2022, 11, 1065  and SoÅ‚tysik and al. BMJ Open 2019;9:e025905. doi:10.1136/ bmjopen-2018-025905. This information must be underlined, better explaining what of new (i.e. data, relationships, elaborations) is reported in this specific new paper respect to the previous ones. Only the negative effects of PA are reported “Both prolonged or short-duration high intensity exercise results in an increased production of reactive oxygen species (ROS) in active skeletal muscles, which results in the formation of oxidized lipids and proteins in the working muscles.” The beneficial effects of physical activity on antioxidant status and oxidative stress, especially in aged subjects, according to the type and intensity of exercise is not reported (Antioxidants 2019, 8, 431; doi:10.3390/antiox8100431).

2. Materials and Methods: 

Often the authors in the text refer to the other/previously published manuscript (i.e Ref 25,26… ) regarding the study design, the protocol, etc. It is an uncomfortable task for the reader to come back into the previous reports in order to find the information. It is suggested to insert in the text some brief information and/or the summary diagram.

Were there any exclusion criteria? Insert please

Year and CE number?

Paragraph 2.1 and 2.2: Briefly explain the protocol

3.Results

The authors report that “In laboratory tests, men had lower number of blood platelets, total cholesterol, LDL-C, HDL-C, higher levels of glucose, uric acid, homocysteine, lower TOS and higher concentration of free thiol groups of plasma proteins……”

Given that the men were significantly older, it could be assumed that they will also take more drugs, as they were suffering from cardiovascular diseases/hypertension/metabolic syndrome, and for this reason, they had lower number of blood platelets, total cholesterol, LDL-C, HDL-C. Please explain and clarify as drug therapies may interact with the oxidative and antioxidative results. 

Moreover is the significantly lower PA-EE reported in men linked to their significantly older age and pathologies?

It is recommended to insert a figure/s with graphics that can help in the immediate understanding of the results.

4.Discussion

Check the references in all text: i.e line 239 “There are results in the literature showing the relationship between PA and antioxidant potential”. Insert references.

Some sentences:

“TAS in bivariate analyzes was influenced by TC and LDL-C. In multivariate model, only LDL-C influenced the TAS (OR:-0.06; CI 95%-0.11--0.02; p<0.001).

In simple correlations, TOS was related to BMI, WHtR, LDL-C and TG. Women were characterized by higher TOS compared to men. This data was presented in previously published work (Table 1) [26]).

None of the included cardiovascular diseases or smoking did affect TOS. The use of ARB was associated with a lower TOS, reversely to the diagnosis of obesity.

Similar to univariate analyses, in the multivariable model, plasma lipid peroxides concentration was influenced by ICAM and presence of stroke [26].

Subjects taking antiplatelet drugs presented significantly lower level of free thiol groups of platelet proteins, similarly to people diagnosed with lipid disorders.”

reported analysis of some parameters that sometimes were already previously reported [26] and that are not evaluated and related to PA in the discussion. Please insert the necessary explanations or delete what is go off the point.

Refers to..”If we look at our data regarding the relationship between PA and the concentration of free  thiol and amino groups in platelet proteins, we may conclude that PA can be perceived as a negative factor, with a pro-oxidant effect, because higher PA is associated with a decrease in the concentration of free thiol and amino groups.” One point is how much physical activity they actually did. It would have been necessary to monitor PA with tools capable of recording this activity, for example smartwatch.